# Physiological Efficiency and Adaptability of Greek Indigenous Grapevine Cultivars Under Heat Stress and Elevated CO_2_: Insights into Photosynthetic Dynamics

**DOI:** 10.3390/plants14162518

**Published:** 2025-08-13

**Authors:** Xenophon Venios, Georgios Banilas, Evangelos Beris, Katerina Biniari, Elias Korkas

**Affiliations:** 1Department of Wine, Vine and Beverage Sciences, University of West Attica, 28, Ag. Spyridonos Str., 12243 Athens, Greece; xvenios@uniwa.gr (X.V.); eberis@uniwa.gr (E.B.); 2Laboratory of Viticulture, Department of Crop Science, Agricultural University of Athens, 75, Iera Odos Str., 11855 Athens, Greece; kbiniari@aua.gr

**Keywords:** grapevine, climate change, photosynthetic dynamics, CO_2_ enrichment, heat stress, cultivars, stomatal traits, water use efficiency

## Abstract

This study investigates the impact of climate change on key physiological parameters of Greek indigenous grapevine cultivars (Savvatiano, Muscat, Assyrtiko, Mavrodafni, Moschofilero, and Agiorgitiko), using Sauvignon blanc and Merlot as benchmarks. The aim was to identify genotypes with higher photosynthetic dynamics and water use efficiency (WUE) under heat stress and to examine the role of CO_2_ enrichment in modulating these responses. Gas exchange measurements showed that short-term exposure to elevated CO_2_ (e[CO_2_]) (i.e., 700 ppm) enhanced photosynthesis by 37–64%, 77–89%, and 18–68% under control, moderate, and severe heat-stress regimes (23, 35, and 40 °C), respectively. CO_2_ enrichment also improved WUE by 61–122%, 96–138%, and 11–63%, with the greatest benefits at 30–33 °C, depending on genotype. Cultivars with strong CO_2_-saturated photosynthetic capacity and small stomata, such as Sauvignon blanc and Mavrodafni, showed greater photosynthetic stimulation and WUE improvement from CO_2_ elevation. Stomatal traits influenced photosynthesis under ambient CO_2_ (a[CO_2_]) but not under e[CO_2_]. Of the white varieties examined, Sauvignon blanc and Savvatiano showed the best performance under combined e[CO_2_] and heat stress, while Assyrtiko and Muscat adapted better to high temperatures at a[CO_2_]. Among red cultivars, Mavrodafni showed the highest photosynthetic efficiency at both CO_2_ conditions, even under heat stress. The present findings indicate that grapevine varieties exhibit differential responses to elevated temperature and CO_2_ levels. A comprehensive understanding of grapevine responses to stress conditions is therefore essential for the selection of cultivars with enhanced adaptation to climate change.

## 1. Introduction

Climate change has received major dimensions in recent years, intensifying the widespread attention and concern of the international community [1]. The dramatic increase in CO_2_ concentration is expected to enhance the greenhouse effect, amplifying the intensity, frequency, and duration of extreme heat events in the future [2].

Given this scenario, major devastating consequences are expected to affect viticulture worldwide, which could be addressed by redefining suitable wine-growing regions, the implementation of appropriate viticultural practices, or the cultivation of heat-resistant cultivars [3,4]. Indeed, although grapevine (*Vitis vinifera* L.) is generally well adapted to dry and hot environments [5], the effects of high temperatures are already evident in many wine-growing regions worldwide, where the summer temperatures reach or surpass 40 °C, affecting growth and yield, with serious risks for viticulture [6]. However, the high degree of grapevine genetic diversity offers a valuable opportunity for exploring a broad genetic spectrum and identifying clones and genotypes capable of thriving in warm wine regions [7,8]. Winegrape cultivars adapted to hotter, drier regions often exhibit better stress responses, higher photosynthetic rates, and consequently yield and quality [9]. This is vital for Mediterranean viticultural zones, characterized by hot and dry summers, where research and identification of heat-adapted cultivars will ensure the sustainability and productivity of viticulture, with significant efforts already being made in this direction [10,11,12].

Greece has a long viticultural tradition and a reconstructive approach to its vineyard. It is expected to play a role in the future of Mediterranean viticulture under climate change. Most viticultural regions in Greece are considered warm, including areas where viticulture might be challenging in the future. However, there are indigenous varieties highly adapted to semi-arid climates, which may contribute to the sustainability of viticulture in the Mediterranean basin [13]. Scientists have already highlighted the imperative need for evaluating the heat-adaptation properties of native cultivars [14,15]. Nevertheless, relatively few studies have dealt thoroughly with the responses of native Greek cultivars to the effects of climate change [16], and almost none have examined their responses under elevated CO_2_ conditions. Notably, existing research has mainly focused on traits such as leaf pigment content, specific leaf structure adaptations, and gene expression of phenolic or aromatic compounds. However, fundamental physiological measurements have not been reported as yet, creating significant knowledge gaps regarding their physiological behavior.

An essential initial step in this direction would be the assessment of Greek cultivars’ photosynthetic performance, which is considered one of the most reliable indicators of heat adaptability. Photosynthesis in grapevine is generally negatively affected by temperatures above 35 °C and is the most heat-sensitive physiological process that can be inhibited before other heat-stress symptoms are detected [17,18,19]. This is largely attributed to biochemical factors such as the attenuation of electron transport rate as well as the inactivation of ribulose-1.5-bisphosphate carboxylase/oxygenase (Rubisco), which catalyses the carboxylation reaction [20]. The biochemical parameters of Vcmax (maximum rate of carboxylation of Rubisco) and Jmax (maximum rate of electron transfer) as temperature-sensitive indicators of photosynthetic capacity can effectively describe the grapevine’s performance under rising temperatures, providing a comprehensive view of heat-stress effects on photosynthetic dynamics [21,22]. Therefore, the assessment of cultivars’ photosynthetic performance may serve as one of the most reliable indicators of heat adaptability.

In addition to biochemical factors, photosynthesis is also limited by stomata regulations, with several reports providing evidence of a tight correlation between heat adaptability and leaf evaporative cooling in grapevine [8,12]. Considering that heat stress is often combined with a high vapour pressure deficit (VPD), there is a shift in the hydraulic balance toward transpiration demand that triggers stomatal closure as a protective mechanism to reduce water loss and prevent xylem cavitation [23,24]. Under water-demanding conditions, optimizing the balance between carbon gain and water consumption, known as intrinsic water use efficiency (iWUE = An/gs), is crucial for maximizing growth and productivity while minimizing water loss through transpiration [25,26]. Consequently, the grapevine’s performance under heat stress is largely determined by its stomatal sensitivity to high VPD conditions. Isohydric cultivars exhibit a more responsive stomata closure under high VPD conditions, providing an effective water conservation strategy, whereas anisohydric cultivars display a reduced stomatal sensitivity to high VPD (higher transpiration), which is considered a mechanism to mitigate heat stress damage [8,27].

The ultimate goal of this study was to compare Greek cultivars’ adaptability under the prevailing high-temperature climate and also to assess their dynamics under a future elevated CO_2_ scenario. The varieties selected for this study were chosen based on their high viticultural and oenological potential, as well as their origin from the southern mainland and insular regions of Greece, a plausible indication of increased adaptation to dry-thermal conditions, with a potential contribution to addressing climate change resilience. Within this framework, we expected to distinguish Greek cultivars with high adaptability under the warm Mediterranean climate conditions, which could potentially replace other heat-sensitive cultivars in the wider viticultural zone, currently or in the near future.

## 2. Results

### 2.1. Physiological Responses Under Ambient CO_2_

Photosynthetic responses (An_400_), stomatal responses (gs_400_), and water use efficiency (WUE_400_) under a[CO_2_] varied greatly among cultivars and temperature treatments (*p* < 0.001). Within the white cultivars, An_400_ revealed thermal optima (Topt) at 37.5–39.5 °C, with Assyrtiko exhibiting the highest An_400_ at 40 °C, while Muscat showed the lowest values over the widest temperature range (Figure 1A). Gs_400_ revealed a positive effect up to 35–37 °C and a downward trend at higher temperatures. Assyrtiko exhibited the highest gs_400_ at 40 °C, while Muscat showed the lowest values over the widest temperature range (Figure 1C). Changes in water use efficiency (WUE_400_) followed cubic curvilinear patterns (R^2^ = 0.972–0.992, *p* < 0.001) that were negatively correlated with temperatures up to 35 °C, followed by a more gradual decline up to 40 °C. Assyrtiko and Sauvignon blanc displayed the highest and the lowest performance, respectively, at 40 °C (*p* < 0.001) (Figure 1E). Within the group of red cultivars, An_400_, gs_400_, and WUE_400_ varied greatly among heat treatments and genotypes (*p* < 0.001). An_400_ revealed Topt at 32–34 °C with Mavrodafni demonstrating the highest An_400_ (*p* < 0.001), and only minor differences were observed among the rest genotypes (Figure 1B). Gs_400_ responses revealed Topt at 35 °C with Mavrodafni demonstrating the best performance across all treatments (*p* < 0.001) (Figure 1D). Mavrodafni displayed the lowest WUE_400_ (2.2–11% lower than the rest), while Moschofilero and Agiorgitiko had the highest values at 35 °C and 40 °C (Figure 1F).

### 2.2. Physiological Responses Under Elevated CO_2_

Photosynthetic responses (An_700_), stomatal responses (gs_700_), and water use efficiency (WUE_700_) under e[CO_2_] varied greatly across cultivars and heat treatments (*p* < 0.001). An_700_ revealed Topt at 35–37.5 °C with Sauvignon blanc recording the highest rates across all temperatures, while Muscat and Assyrtiko were 25–28% lower at 40 °C (*p* < 0.001) (Figure 2A). Gs_700_ revealed Topt at 35.5–38.5 °C with Sauvignon blanc exhibiting the highest values, while Muscat showed the lowest over the widest temperature range (*p* < 0.001) (Figure 2C). The temperature dependence of WUE_700_ followed cubic curvilinear patterns (R^2^ = 0.98–0.996, *p* < 0.001) that were negatively correlated with temperature. Sauvignon blanc exhibited the highest and Muscat the lowest WUE_700_ across treatments (Figure 2E). Among red cultivars, significant variations were also observed in An_700_, gs_700_, and WUE_700_ across all the heat treatments (*p* < 0.001). An_700_ revealed Topt at 33–34 °C with Mavrodafni exhibiting the highest rates across all temperatures (*p* < 0.001), whereas only minor differences were observed among the rest genotypes (Figure 2B). Gs_700_ revealed Topt at 35–36 °C with Mavrodafni displaying the highest values (*p* < 0.001) and Moschofilero significantly the lowest across all temperatures (Figure 2D). Mavrodafni also presented the highest WUE_700_, while Agiorgitiko and Merlot showed the lowest values across treatments (*p* < 0.001) (Figure 2F).

### 2.3. Physiological Responses to CO_2_ Elevation

Ιn both varietal groups, the results revealed a substantial photosynthetic stimulation by CO_2_ enrichment with great variation among genotypes that was more pronounced at marginal temperatures (23 °C and 40 °C) (*p* < 0.001). Growth temperature changes significantly affected the degree of photosynthetic stimulation, which peaked at 33–35 °C, depending on genotype. The impact of CO_2_ elevation on stomatal responses also varied significantly among heat treatments and genotypes (*p* < 0.001) in both varietal groups. Water use efficiency (WUE) estimates revealed a positive effect of CO_2_ enrichment, highly dependent on temperature in a cubic relationship (R^2^ = 0.98–0.99, *p* < 0.001). Topt occurred at 30–33 °C, depending on genotype, in both varietal groups (Figure 3).

### 2.4. Temperature Dependence of Photosynthetic Capacity Parameters

Temperature responses of Vcmax (R^2^ = 0.976–0.982, *p* < 0.001) and Jmax (R^2^ = 0.883–0.982, *p* < 0.001) followed different curvilinear patterns, revealing different Topt (Vcmax = 38.4–39.5 °C vs. Jmax = 35.8–37.2 °C). Vcmax and Jmax varied greatly among treatments and genotypes (*p* < 0.01 and *p* < 0.001, respectively) in both varietal groups. Among white cultivars, Sauvignon blanc consistently presented the highest Vcmax across treatments, while Muscat showed 25–31% lower rates (Figure 4A). Sauvignon blanc consistently presented the highest Jmax across treatments, while Muscat exhibited 23–29% lower rates (*p* < 0.001) (Figure 4C). Mavrodafni exhibited the highest Vcmax among red cultivars (*p* < 0.005), while Merlot showed the lowest rates (*p* < 0.025) (Figure 4B). Mavrodafni exhibited the highest Jmax (*p* < 0.013), while Merlot showed rates reduced by 28.5–30.4% (*p* < 0.001) (Figure 4D).

The relationship between Jmax and Vcmax showed a strong linear correlation (R^2^ = 0.706, *p* < 0.001). The activation energy was higher for Vcmax compared to Jmax (45.8–65.5 versus 19.6–32.7 kJ mol^−1^, respectively), resulting in an exponential decrease in the ratio Jmax/Vcmax as the temperature rose up to 40 °C. No significant differences in Jmax/Vcmax were observed between genotypes across treatments for both varietal groups (*p* > 0.059) (Figure 5A,B). The CO_2_-saturated photosynthesis (Amax) was also affected by rising temperature, with values being almost doubled compared to An_400_. The temperature responses revealed Topt at 34.8–35.6 °C for the white cultivars and 34.1–34.8 °C for the red ones (Figure 5C,D).

### 2.5. Stomatal, Mesophyll, and Biochemical Limitations of Photosynthesis

Stomatal limitations (Ls) revealed a negative non-linear relationship with temperature (R^2^ = 0.275, *p* = 0.002), with significant variations between cultivars at 35 °C and 40 °C but not at 23 °C for the white genotypes (*p* < 0.05). Mesophyll limitations (Lm) were minimally affected by growth temperature changes with significant variations among genotypes at 35 °C and 40 °C but not at 23 °C (*p* < 0.001). Biochemical limitations (Lb) significantly increased with rising temperature, showing significant variations between cultivars at 35 °C and 40 °C but not at 23 °C for the white genotypes (*p* < 0.006) (Figure 6).

### 2.6. Morphological Leaf Traits Contributing to Physiological Responses

Stomatal morphology in terms of size, density, and stomatal index varied greatly across cultivars and treatments (*p* < 0.001). Muscat had the largest stomata, while Sauvignon blanc, along with Assyrtiko, had the smallest ones, reduced by 29–35% (*p* < 0.001) (Figure 7A). Assyrtiko displayed the highest stomatal density and Muscat significantly the lowest among white cultivars (*p* < 0.001) (Figure 7C). Assyrtiko showed the highest stomatal index (SI) at 40 °C (*p* < 0.001), while Muscat showed the lowest value (Figure 7E). Among red cultivars, Agiorgitiko had the largest stomata, while Mavrodafni had the smallest ones, reduced by 26.6% (*p* < 0.001) (Figure 7B). Mavrodafni exhibited the highest density (*p* < 0.001), while Agiorgitiko had the lowest (*p* < 0.003), with values approximately 47% lower (Figure 7D). Mavrodafni also demonstrated the highest SI (*p* < 0.001), while no significant differences were observed among the rest genotypes (Figure 7F). Α strong negative correlation was revealed between stomatal size and density (r = 0.690, *p* < 0.001).

## 3. Discussion

### 3.1. Biochemical Contribution in Differential Photosynthetic Dynamics Under Elevated CO_2_

Many studies have revealed that increased photosynthesis under elevated CO_2_ is mainly due to the increased activity of Rubisco (Ribulose 1,5-bisphosphate carboxylase-oxygenase), the catalyst of RuBP (Ribulose 1,5-bisphosphate) carboxylation, which is a required reaction for CO_2_ fixation [28]. Therefore, an increase in atmospheric CO_2_ levels also raises the CO_2_ concentration surrounding Rubisco, leading to higher carboxylation rates and, consequently, enhanced photosynthetic activity [29]. Our study demonstrated that short-term acclimation to e[CO_2_] had a profound stimulatory effect on photosynthesis, with increasing rates by 37–64%, 77–89%, and 18–68% under control, moderate, and severe heat stress regimes, respectively. Similar findings have been reported in the grapevine cultivars Touriga Franca [30], Shiraz [31], and Tempranillo [32]. However, the imposition of heat stress tends to diminish this photosynthetic enhancement at temperatures above 35 °C [33]. For instance, in Sangiovese, the 40–45% yield stimulation under e[CO_2_] was reduced or eliminated at high temperatures [34]. Thus, although elevated CO_2_ typically enhances photosynthesis [35,36,37], rising temperatures can positively or negatively affect this stimulation, depending on whether temperature is below or above the thermal optimum [38,39,40]. Previous studies have associated the reduced benefit from CO_2_ enrichment with Jmax decreases at high temperatures [41,42]. This is consistent with the fact that e[CO_2_] shifts the limitation of photosynthesis from predominantly Rubisco activity to predominantly RuBP regeneration capacity [43]. Therefore, under e[CO_2_], reductions in Vcmax have a minor effect on photosynthesis, whereas decreases in Jmax impose a proportionally greater limitation [37]. Indeed, our results, in agreement with findings in Syrah and Grenache [22], showed a significant linear correlation between An_700_ or An boost with Jmax but not with Vcmax. In our study, all cultivars exhibited significant reductions in An_700_, An boost, and Jmax above 35 °C. These reductions were more pronounced in Assyrtiko, suggesting a lower CO_2_-saturated photosynthetic capacity under heat stress. Among the red cultivars, Mavrodafni exhibited the strongest photosynthetic response to e[CO_2_], whereas Merlot benefited the least due to its inherently low Jmax values.

### 3.2. The Role of Stomata Traits in Modulating Photosynthesis Under Elevated CO_2_

Elevated CO_2_ triggers stomatal closure via an ABA-independent signaling pathway, while basal ABA signaling and OST1/SnRK2 activation are still required to amplify the response [44]. Our results revealed a CO_2_-induced stomatal closure up to 26.4% for the red cultivars and up to 27.3% for the white cultivars, which is consistent with other grapevine studies [45,46]. However, e[CO_2_] did not induce stomatal closure in all cases; instead, a tendency for reduced stomatal closure or even opening was observed at high temperatures and increased VPD. These observations are in agreement with studies conducted on various woody tree and shrub species [47] as well as on grapevine cultivars such as Riesling and Cabernet Sauvignon [48]. The latter reported that grapevine plants can respond to e[CO_2_] by increasing their gs under hot and dry atmospheric conditions. Our study identified a positive correlation between stomatal size and gs, suggesting that cultivars with larger stomata (i.e., Muscat and Agiorgitiko) exhibit less hermetic stomatal closure under e[CO_2_]. Previous studies have also reported that e[CO_2_] may favour plants with larger stomata, which maintain higher gs and improved carbon uptake, whereas a[CO^2^] favours species with smaller stomata [49,50]. In the present study, although e[CO_2_] induced stomatal closure, An increased, a fact that, according to Rosenthal et al. [42], is attributable to the nonlinear correlation between gs and ci. Additionally, An_700_ was more correlated with Jmax than gs_700_, suggesting a rather minimal role of stomata in regulating photosynthesis under e[CO_2_]. Similar findings in *Populus tomentosa* and *Eucalyptus robusta* were attributed to the fact that e[CO_2_] shifts the predominant limitation on photosynthesis from stomatal to biochemical components, particularly electron transport rate (Jmax) [51]. Indeed, our results confirmed that cultivars with predominantly stomatal-limited photosynthesis (Sauvignon blanc, Savvatiano, Mavrodafni) exhibited a photosynthetic advantage under e[CO_2_], whereas cultivars with more biochemically-limited photosynthesis (Assyrtiko, Muscat, Merlot) performed better under a[CO_2_].

### 3.3. WUE Benefits from CO_2_ Elevation

In addition to An boost, CO_2_ elevation minimized water loss by lowering stomatal conductance and enhancing iWUE with similar results also in Cabernet Sauvignon [48,52], Tempranillo [32], and Riesling [48]. Considering that iWUE is defined as An/gs, the magnitude of WUE improvement under e[CO_2_] depends on the relative balance between An stimulation and stomatal closure. Our results revealed a great temperature effect on this benefit with Topt at 30–33 °C where the ratio of An stimulation/stomatal closure was the highest. WUE improvement can also vary between grapevine cultivars, as reported in Cabernet Sauvignon and Chardonnay, where the former exhibited up to 20% higher WUE in response to e[CO_2_] [53]. In our study, the reduced WUE improvement in Muscat and Agiorgitiko was attributed to their large, less responsive to e[CO_2_] stomata. In Assyrtiko and Merlot, the diminished An stimulation above 35 °C resulted in a lower WUE benefit from e[CO_2_] under heat stress. In contrast, Sauvignon blanc and Mavrodafni exhibited a remarkable WUE improvement from e[CO_2_], driven by a combination of a strong An stimulation and a hermetic CO_2_-induced stomatal closure. These data suggest that cultivars with smaller stomata and higher Jmax exhibit a greater WUE benefit from e[CO_2_] due to their aggressive stomata closure and better photosynthetic responsiveness to e[CO_2_]. Regarding stomatal responsiveness to e[CO_2_], findings in *Arabidopsis thaliana* confirm that plants with larger stomata exhibited a comparatively higher gs when exposed to e[CO_2_] and therefore lower WUE benefits [49]. Other studies also suggest that smaller stomata respond faster under dynamic environments such as changing CO_2_ concentration, due to a higher surface area to volume ratio [54,55,56]. This trait enables gs to increase or decrease more rapidly, leading to a better balance between CO_2_ uptake and water loss, and therefore higher WUE benefits from CO_2_ elevation.

### 3.4. Biochemical Contribution in Photosynthetic Variations Under Ambient CO_2_

In contrast to e[CO_2_], where RuBP is saturated and photosynthesis is limited by RuBP regeneration (Aj), a[CO_2_] RuBP is unsaturated and RuBP carboxylation (Ac) is the main biochemical limiting factor of photosynthesis [57]. In fact, at a[CO_2_], there is often a rebalancing between these two processes with rising temperature, in favour of Vcmax at higher temperatures, which is reflected in a lower Jmax to Vcmax ratio [58,59,60]. In grapevine, the transition point from RuBP regeneration to RuBP carboxylation-limited photosynthesis was found to occur at 30 °C and 35 °C, in Semillon [19] and Chardonnay [61], respectively. Therefore, photosynthetic reductions at temperatures above 35 °C can be mainly explained by Vcmax rather than Jmax [62]. Sauvignon blanc and Savvatiano demonstrated high carboxylation capacities, providing a significant biochemical advantage over Assyrtiko and Muscat, whose photosynthesis was mainly carboxylation-limited. Mavrodafni also demonstrated a high carboxylation efficiency under heat stress in contrast to Merlot, whose photosynthesis was predominantly carboxylation-limited. The observed negative correlation between Vcmax and mesophyll limitations indicates that the restricted CO_2_ diffusion through mesophyll to the sites of carboxylation significantly constrains carboxylation capacity, contributing to cultivar-specific photosynthetic performance under heat stress conditions.

### 3.5. Stomatal Strategies and Cultivar Adaptation to Heat Stress Under Ambient CO_2_

The present results revealed a substantial increase in gs from 23 °C to 35 °C, especially in Mavrodafni and Sauvignon blanc, which is consistent with other grapevine cultivars such as Sangiovese [63] and Merlot [64]. However, temperatures above 35 °C and VPD > 3 Kpa induced stomatal closure, due to extreme evaporative demands, leading to a decline in photosynthesis, which is a common response reported in grapevine cultivars such as Semillon, Chardonnay, Campbell Early, and Kyoho [61,65,66,67]. A comparison of gs_400_ revealed that Sauvignon blanc and Savvatiano exhibited similar stomatal responses under a[CO_2_], both peaking at 35 °C, and with comparable declines in slope at higher temperatures. In contrast, Assyrtiko and Muscat exhibited a higher Topt (36–37 °C) with a better retention of opened stomata above 35 °C, resulting in a stronger tendency to maintain high photosynthetic rates up to 40 °C. These results suggest that both cultivars exhibit a remarkable heat stress adaptability and improved capacity to cope with extreme evaporative demands (high VPD) by adopting anisohydric-like stomatal responses similar to Semillon [68] and Shiraz [69]. Isohydricity or anisohydricity in grapevine is associated with anatomical/hydraulic differences as well as distinct genetic and hormonal responses, particularly involving ABA regulation within the xylem sap [70], which reflect the adaptation of cultivars to different soil and climate conditions [71]. The aforementioned offensive stomatal strategy might be inextricably linked to the adaptation of these cultivars to the high VPD conditions typical of insular Greece, as Muscat originates from Samos and Assyrtiko from Santorini. Conversely, Sauvignon blanc, typically cultivated in cooler regions, might not have developed such adaptive mechanisms to cope with hot and dry climatic conditions. It exhibited increased stomatal sensitivity to VPD, involving a tighter stomatal closure to mitigate the effects of extreme evaporative demands, similar to isohydric grapevine cultivars such as Grenache [70] and Cabernet Sauvignon [72]. Interestingly, although Savvatiano originates form Attica, a relatively hot and dry region, it also adopts a similar strategy of closing the stomata under high VPD conditions to prevent excessive water loss.

### 3.6. The Role of Stomata Traits in Modulating Photosynthesis Under Ambient CO_2_

By investigating the role of stomatal traits in determining photosynthesis under a[CO_2_], the Pearson coefficient revealed a strong negative correlation between stomatal size and An_400_ and a positive correlation between stomatal density and An_400_. Although there are a lack of respective data in grapevine, these results are in accordance with findings in *Arabidopsis thaliana* [73], *Solanum lycopersicum* [74], *Oryza sativa* L. [75], *Banksia* spp. [55], and several woody plant species [76], where plants with small and many stomata tended to record higher photosynthetic rates. Indeed, Assyrtiko and Sauvignon blanc showed a better photosynthetic performance than Muscat, while Mavrodafni, having small and numerous stomata, exhibited the highest photosynthesis among red cultivars. We also detected a positive correlation between stomatal index and An_400_, confirming the findings of Segev et al. [77] that higher stomatal indices are associated with higher photosynthetic rates in a non-water stressed environment. This may explain the higher photosynthesis of Assyrtiko and Mavrodafni under [CO_2_]-heat stress conditions as compared to other cultivars, possibly due to their high stomatal indices.

## 4. Materials and Methods

### 4.1. Plant Material and Growth Conditions

Eight *Vitis vinifera* cultivars were used, categorized into two groups: the white Greek cultivars Savvatiano, Assyrtiko, and Muscat (also known as Muscat of Samos) and the red Greek cultivars Moschofilero, Agiorgitiko, and Mavrodafni. The white and red cultivars Sauvignon blanc and Merlot, respectively, were also included in the analysis for comparison reasons. Vines were 2 years old grafted onto Richter 110 rootstock and grown outdoors in 8lt pots (peat 60%, clay 30%, and zeolite 10%). They were fertilized regularly (Nutri-Leaf 20-20-20 + T.E., Miller, Hanover, PA, USA) and irrigated daily to saturation. After budbreak, vines were pruned to maintain one shoot. When 10–11 fully expanded leaves developed, they were transferred to a growth chamber (CMP-6050, Conviron, Winnipeg, MB, Canada) equipped with T5 fluorescent and halogen lamps and a CO_2_ additive control package including a gas analyzer for regulating the CO_2_ concentration. The lamp operation was set to a constant light intensity of 1000 μmol m^−2^ s^−1^ into the growth chamber, the day/night photoperiod cycle was set at 13/11 h, and the relative humidity was maintained between 50 and 60%. Plants were subjected to three temperature regimes (day/night): control (CT) at 23 °C/15 °C, moderate heat stress (MHS) at 35 °C/27 °C, and severe heat stress (SHS) at 40 °C/32 °C. Two CO_2_ concentrations were tested: 400 ppm (ambient) and 700 ppm (elevated). Surveys were conducted consecutively using a single growth chamber (Conviron CMP-6050). Each group of cultivars (white or red) was separately placed into the chamber and consisted of a total of 16 plants, corresponding to four plants per variety (*n* = 4). Each group of cultivars was initially exposed to CT, then to MHS, and finally to SHS, with 400 ppm applied first and 700 ppm thereafter at each temperature regime. After each transition from one condition to another, the plants were acclimatized for 3–4 days upon the initiation of measurements.

### 4.2. Gas Exchange Measurements

Leaf gas exchange measurements were conducted using a portable photosynthesis system (LI-COR 6400XT, Li-Cor, Lincoln, NE, USA) equipped with a 6 cm^2^ leaf chamber. The CO_2_ concentration into the leaf chamber was monitored by an infrared CO_2_ analyzer and set to match the growth CO_2_ level applied (400 or 700 ppm). Similarly, the leaf temperature was set equal to growth temperature (23 °C, 35 °C or 40 °C). The light intensity, which was regulated by an LED light source (6400-02B, LI-COR Biosciences, Lincoln, NE, USA) attached to the LI-6400XT sensor head, was set at 1000 μmol m^−2^ s^−1^. All measurements were conducted on mature fully expanded leaves. Prior to measurement, plants were acclimatized for 4–5 days under each condition.

### 4.3. Rapid A-Ci Response Curves (RACiR)

The RACiR curve method was based on Lawrence et al. [78]. All the curves were performed on the 8th full-expanded leaf from the base, with a photon flux density (PFD) of 1000 μmol m^−2^ s^−1^, relative humidity of 50–60%, and ambient CO_2_ concentration (400 ppm) at control (23 °C), moderate (35 °C) and severe heat stress (40 °C) temperature regimes.

### 4.4. Biochemical Model Description

Data obtained from the Rapid A-Ci curves were fitted into the Ethier’s and Livingston [79] photosynthetic model for the determination of a maximum rate of carboxylation (Vcmax) and maximum rate of electron transport (Jmax). Non-linear least-squares regression was applied for curve fitting using SPSS Statistics v29. According to this model, the photosynthetic rate (A) is the minimum of two limiting processes:(1)A=min(Ac,Aj),
where Ac (μmol m^−2^ s^−1^) is the rate of photosynthesis limited by RuBP carboxylation and Aj (μmol m^−2^ s^−1^) is the rate of photosynthesis limited by RuBP regeneration through the electron transport rate. Under Rubisco-limited conditions, the response of photosynthesis to CO_2_ is a function of the kinetic properties of Rubisco (K*_c_*, K*_o_*, Γ*) and C*_c_* given from the following equation:(2)$$\mathrm{Ac}=\frac{\mathrm{Vc}\max\left(\mathrm{Cc}-\Gamma^{*}\right)}{\mathrm{Cc}+\mathrm{Kc}\left(1+\frac{\mathrm{Oi}}{\mathrm{Ko}}\right)}-\mathrm{Rd},$$
where Vcmax is the maximum activity of Rubisco, Ci and Oi are the intercellular concentrations of CO_2_ and O_2_, respectively, Kc and Ko are the Michaelis constants for CO_2_ and O_2_, respectively, Rd is the mitochondrial respiration rate, and Γ* is the chloroplast CO_2_ compensation point in the absence of mitochondrial respiration (Rd). Similarly, when photosynthesis is limited by RuBP regeneration through electron transport, the photosynthetic rate is given from the following equation:(3)$$\mathrm{Aj}=\frac{J\left(\mathrm{Cc}-\Gamma^{*}\right)}{4\mathrm{Cc}+8\Gamma^{*}}-\mathrm{Rd},$$
where J is the rate of electron transport for a given PFD. A non-rectangular hyperbolic function was used to relate J to PFD, as follows:(4)$$\Theta J^{2}-\left(\mathrm{aPFD}+J\max\right)J+\mathrm{aPFD}*J\max=0,$$
where Jmax is the potential maximum rate of electron transport, Θ is the curvature of the photosynthetic light response curve, and α is the photon yield of electron transport.

### 4.5. Mesophyll Conductance Estimation

Mesophyll conductance (gm) was determined by the variable J method using combined chlorophyll fluorescence and gas exchange measurements as follows [80]:(5)$$\mathrm{ETR}=\left(\mathrm{An}+\mathrm{Rd}\right)\frac{(\mathrm{Ci}-\mathrm{An}/\mathrm{gm})+2\Gamma^{*}}{(\mathrm{Ci}-\mathrm{An}/\mathrm{gm})-\Gamma^{*}},$$
where ETR was obtained from chlorophyll fluorescence measurements, and An and Ci were obtained from gas exchange measurements at 1000 μmol m^−2^ s^−1^. Γ* was taken after Bernacchi et al. [81], and dark respiration was used as a proxy for Rd [82].

Estimated gm values were then used to convert substomatal CO_2_ concentration (Ci) into chloroplast CO_2_ concentration (Cc) using the following equation [32]:(6)$$\mathrm{Cc}=\mathrm{Ci}-\frac{\mathrm{An}}{\mathrm{gm}},$$

### 4.6. Stomatal, Mesophyll, and Biochemical Limitations of Photosynthesis

The stomatal limitation of photosynthesis was calculated from rapid A/Ci curves, using the following equation [83], which describes the diffusion of CO_2_ from the atmosphere to the intercellular spaces as(7)$$\mathrm{Ls}=1-\frac{\mathrm{Aa}-\mathrm{gs}}{\mathrm{Ai}-\mathrm{gs}}*100,$$
where Aa-gs is the rate of photosynthesis (An) at the growth Ca (400 ppm) and Ai-gs is the value of An when Ci equals the growth Ca (Ci = Ca = 400 ppm). The relative limitation to photosynthesis imposed by gm was calculated from the A/Cc response curves as follows [82]:(8)$$\mathrm{Lm}=1-\frac{\mathrm{Aa}-\mathrm{gm}}{\mathrm{Ai}-\mathrm{gm}}*100,$$
where Aa-gm is the value of An estimated graphically using the actual gm, and Ai-gm is the value of An estimated if gm was infinite.

Finally, considering that non-stomatal limitations are defined as the sum of the contributions due to mesophyll conductance and leaf biochemistry, the limitation of photosynthesis attributed to biochemical factors is expressed by the following equation:(9)Lb=100−Ls−Lm,
where Lb = biochemical limitations, Ls = stomatal limitations, and Lm = mesophyll conductance limitations.

### 4.7. Temperature Dependence Models for Curve Estimation

Temperature dependence of Vcmax and Jmax were estimated by fitting the measured values to a temperature-dependence model that is based on two basic functions. The first is the Arrhenius equation, which expresses the rate of a biological reaction below optimum temperature, as follows:(10)$$f(\mathrm{Tk})=K_{23}\exp\left[\frac{\mathrm{Ha}\left(\mathrm{Tk}-296\right)}{296\mathrm{RTk}}\right],$$

The second is a temperature-response formula [84], which is a modified Arrhenius function including a deactivation term that expresses the potential decrease in the rate of a reaction above optimum temperature, as follows:(11)$$f(\mathrm{Tk})=K_{23}\exp\left[\frac{\mathrm{Ha}\left(\mathrm{Tk}-296\right)}{296\mathrm{RTk}}\right]\frac{1+\exp\frac{296\Delta S-\mathrm{Hd}}{296R}}{1+\exp\left(\frac{\mathrm{Tk}\Delta S-\mathrm{Hd}}{\mathrm{TkR}}\right)},$$
where K_23_ is the rate of the metabolic process at Tref = 23 °C (296 K), Ha is the activation energy of the reaction, which represents the rate of the increase in the reaction below the optimum temperature (Topt), Hd represents the rate of the decrease in the reaction above the Topt, and ΔS is an entropy term. Both equations were normalized with respect to 23 °C (296 K) in our experiment, while Hd was set to 200 Kj mol^−^^1^ [85]. Based on the above equations, the optimum temperature (Topt) for a reaction was calculated as follows:(12)$$\mathrm{Topt}=\frac{\mathrm{Ha}}{\Delta S-R\ln\left(\frac{\mathrm{Hd}}{\mathrm{Hd}-\mathrm{Ha}}\right)},$$

Τhe abovementioned equations were also applied for the determination of Topt of photosynthesis (An_400_, An_700_, Amax). The resulting Topt values were then inserted into a Gaussian growth function model [86], which provides bell-shaped curves with great flexibility and accurate fit to the measured data and it is described as follows:(13)$$A(T)=\mathrm{Aopt}*\exp\left(-\frac{{\left(T-\mathrm{Topt}\right)}^{2}}{\sigma^{2}}\right),$$

Topt is derived from the modified Arrhenius function, Aopt represents the maximum photosynthetic rate occurring at Topt, and the σ parameter indicates the spread of the Gaussian curve, reflecting how narrowly or broadly it is distributed around Topt.

### 4.8. Stomatal Traits

Stomatal trait analysis was conducted using healthy, fully expanded leaves of the same size and age and from locations exposed to full sunlight. Samples were prepared for observation by applying clear nail polish to the abaxial leaf surface. Transparent adhesive tape was then applied to the dried surface and removed, resulting in the impression of the leaf surface. Stomatal impressions were observed under an optical microscope (Leica DM1000, Leica Microsystems GmbH, Wetzlar, Germany) and photographed with a digital camera (Leica DFC450). All the stomata and epidermal cells within 1 mm^2^ were counted, taking measurements from 10 squares on the leaf surface, avoiding areas at the edge and near the midrib of the leaf. The photos gained, were then loaded into the image J software (https://imagej.net/ij/ accessed on 3 August 2025) for stomatal size and density determination. The size of stomata was calculated using the following equation [87]:(14)Size=length*∑(widths),
where length is stomatal length and **∑**(widths) is the sum of two guard cell widths. Stomatal index (SI, %) is also calculated from the following equation [88]:(15)$$\mathrm{SI}=\frac{S}{S+E}*100\%,$$
where S is the number of stomata and E is the number of epidermal cells.

### 4.9. Statistical Analysis

Analysis of variance (ANOVA) was conducted to assess the effects of independent variables (variety and temperature) on all dependent variables (An, gs, WUE, Vcmax, Jmax, Amax, Ls, Lm, Lb, stomatal size/density, and SI), followed by Tukey’s HSD post-hoc test. Prior to performing ANOVA, the assumptions of normality and homogeneity of variances were tested using the Shapiro–Wilk and Levene’s tests, respectively, at a significance level of *p* ≤ 0.05. Paired samples *T*-tests were conducted to evaluate the impact of CO_2_ concentration on An, gs, and WUE as the independent variable (CO_2_ concentration) has only two levels (400 ppm, 700 ppm) making the use of ANOVA and Tukey’s HSD post-hoc test inapplicable for this particular comparison. Linear and non-linear regression analyses were also performed for curve normalization, fitting data to 3rd and 4th degree polynomials. Significant differences between means were assessed using Tukey’s multiple comparison test and paired samples *T*-tests at *p* ≤ 0.05. All statistical analyses were conducted using SPSS Statistics v29.

## 5. Conclusions

Heat stress exposure, alone or in combination with e[CO_2_], revealed different photosynthetic dynamics and physiological adaptabilities among cultivars. Within the white group of cultivars, Sauvignon blanc and Savvatiano were the most benefited from CO_2_ enrichment. The best performance under a[CO_2_] and heat stress conditions was recorded for Assyrtiko followed by Muscat, while Sauvignon Blanc and Savvatiano did not respond well to heat stress. Regarding the red cultivars, Mavrodafni exhibited a photosynthetic advantage under e[CO_2_] compared to other cultivars. It also exhibited the best performance under heat stress conditions, regardless of CO_2_ concentration, while Moschofilero, Agiorgitiko, and Merlot displayed significantly lower physiological responses. Although Assyrtiko originates from an Aegean island characterized by a hot and dry climate, there was not a clear link between a cultivar’s performance under heat stress and the specific climate of its region of origin. For instance, Savvatiano, a variety native to Attica, performed similarly to Sauvignon blanc.

The present results highlight the importance of understanding grapevine physiological responses to facilitate the targeted selection of varieties based on their adaptability to specific climatic conditions. Furthermore, the identification of heat-adapted cultivars under both a[CO_2_] and e[CO_2_] conditions is essential for the sustainability and productivity of viticulture in future climate scenarios. Future research should investigate the effects of elevated CO_2_ and heat stress on a broader range of physiological responses, as well as on berry composition and wine quality. In addition, investigating gene expression profiles and hormonal responses could provide deeper insights into the underlying physiological mechanisms.

## Figures and Tables

**Figure 1 plants-14-02518-f001:**
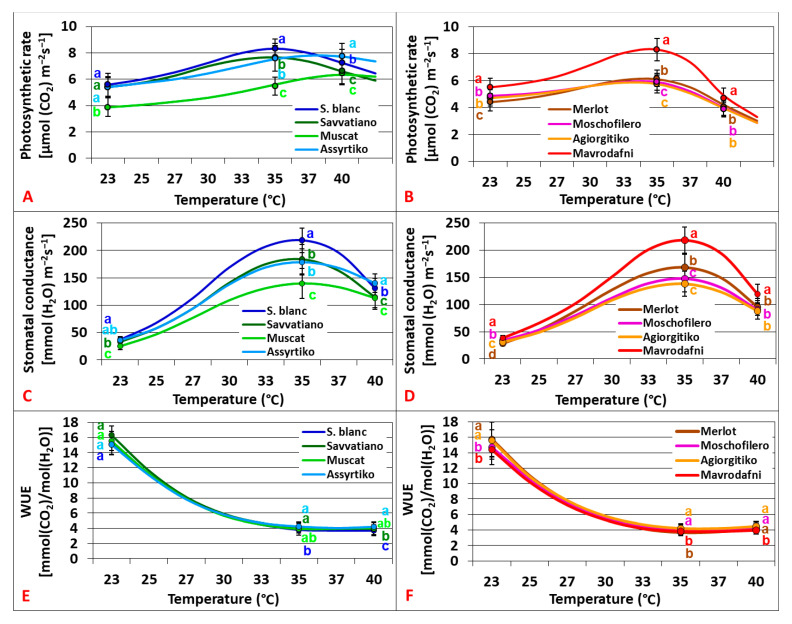
Temperature dependence of photosynthesis (**A**,**B**), stomatal conductance (**C**,**D**), and water use efficiency (**E**,**F**) of grapevine cultivars grown under a[CO_2_] (400 ppm). Data points with error bars represent mean ± standard deviation (SD). Different lowercase letters denote statistically significant differences between cultivars (*p* < 0.05). (**A**,**C**,**E**) refer to white cultivars and (**B**,**D**,**F**) to red ones (*n* = 4).

**Figure 2 plants-14-02518-f002:**
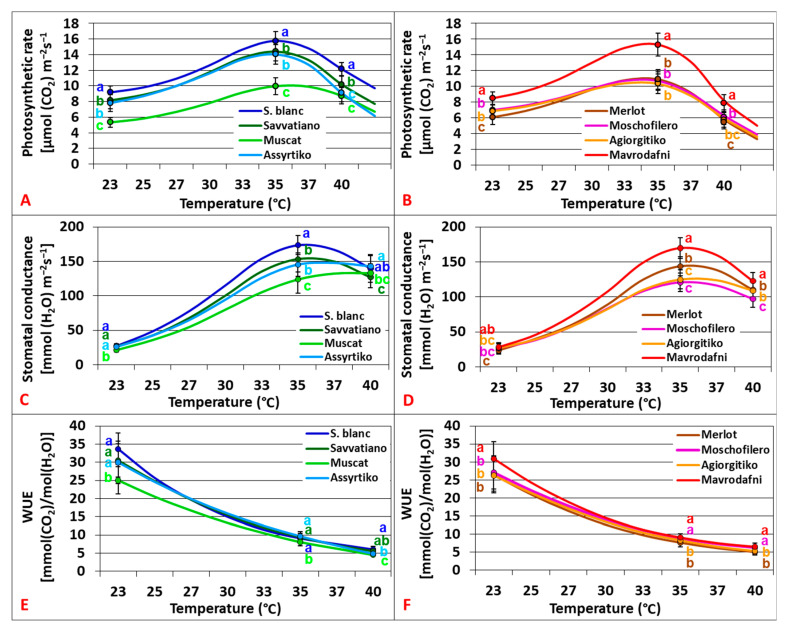
Temperature dependence of photosynthesis (**A**,**B**), stomatal conductance (**C**,**D**), and water use efficiency (**E**,**F**) of grapevine cultivars grown under e[CO_2_] (700 ppm). Data points with error bars represent the mean ± SD. Different lowercase letters denote statistically significant differences between cultivars (*p* < 0.05). (**A**,**C**,**E**) refer to white cultivars and (**B**,**D,F**) to red ones (*n* = 4).

**Figure 3 plants-14-02518-f003:**
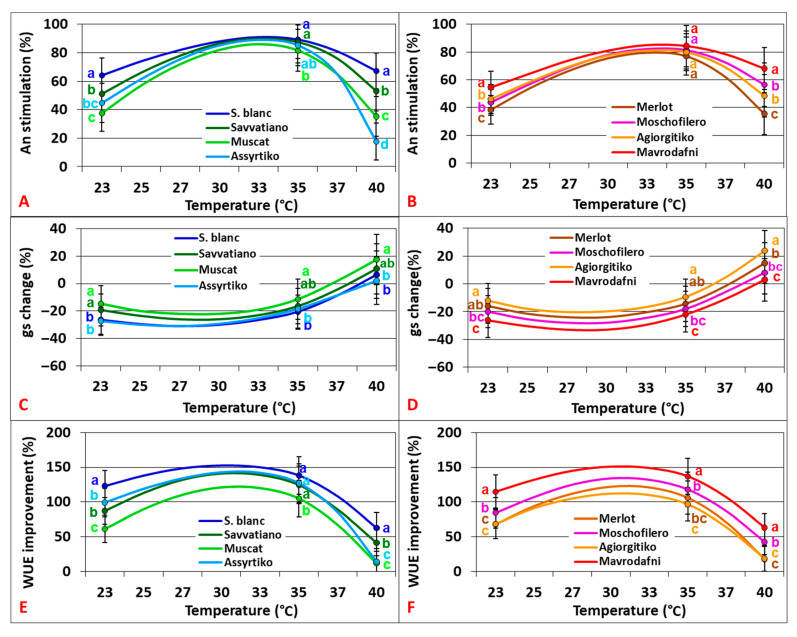
Temperature dependence of photosynthetic (An) stimulation (**A**,**B**), stomatal conductance (gs) change (**C**,**D**), and water use efficiency (WUE) improvement (**E**,**F**) under the effect of CO_2_ elevation (from 400 to 700 ppm) of grapevine cultivars. Data points with error bars represent the mean ± SD and different lowercase letters denote statistically significant differences between cultivars (*p* < 0.05). (**A**,**C,E**) refer to white cultivars and (**B**,**D**,**F**) to red ones (*n* = 4).

**Figure 4 plants-14-02518-f004:**
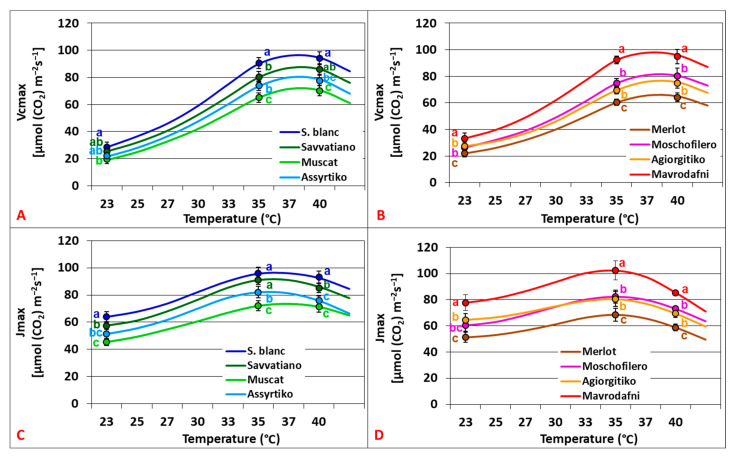
Temperature dependence of maximum carboxylation rates (Vcmax) (**A**,**B**) and maximum electron transport rates (Jmax) (**C**,**D**) of grapevine cultivars. Data points with error bars represent the mean ± SD, and different lowercase letters denote statistically significant differences between cultivars (*p* < 0.05). (**A**,**C**) refer to white cultivars and (**B**,**D**) to red ones (*n* = 4).

**Figure 5 plants-14-02518-f005:**
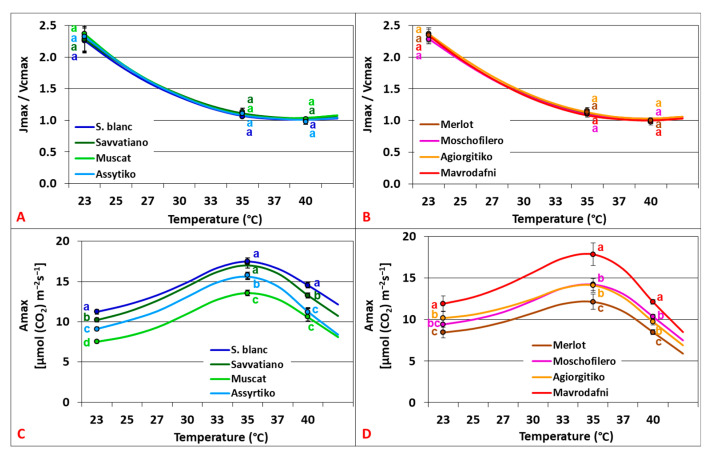
Temperature dependence of the Jmax/Vcmax ratio reflecting the relationship between the two main biochemical processes that limit photosynthesis (RuBP regeneration and RubP carboxylation) (**A**,**B**) and CO_2_-saturated photosynthesis (Amax) (**C**,**D**) of grapevine cultivars. Data points with error bars represent the mean ± SD, and different lowercase letters denote statistically significant differences between cultivars (*p* < 0.05). (**A**,**C**) refer to white cultivars and (**B**,**D**) to red ones (*n* = 4).

**Figure 6 plants-14-02518-f006:**
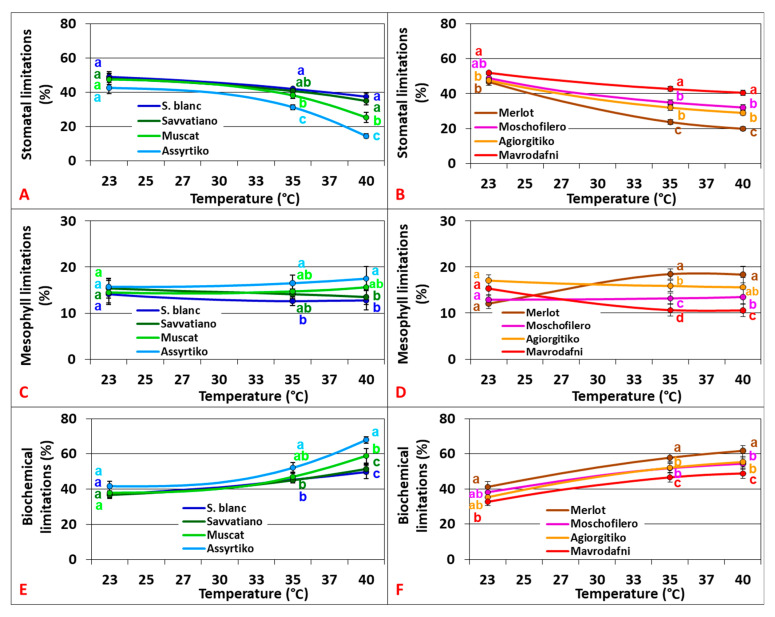
Stomatal (**A**,**B**), mesophyll (**C**,**D**), and biochemical limitations (**E**,**F**) of photosynthesis for the tested grapevine cultivars. Data points with error bars represent the mean ± standard deviation, and different lowercase letters denote statistically significant differences between cultivars (*p* < 0.05). (**A**,**C**,**E**) refer to white cultivars and (**B**,**D**,**F**) to red ones (*n* = 4).

**Figure 7 plants-14-02518-f007:**
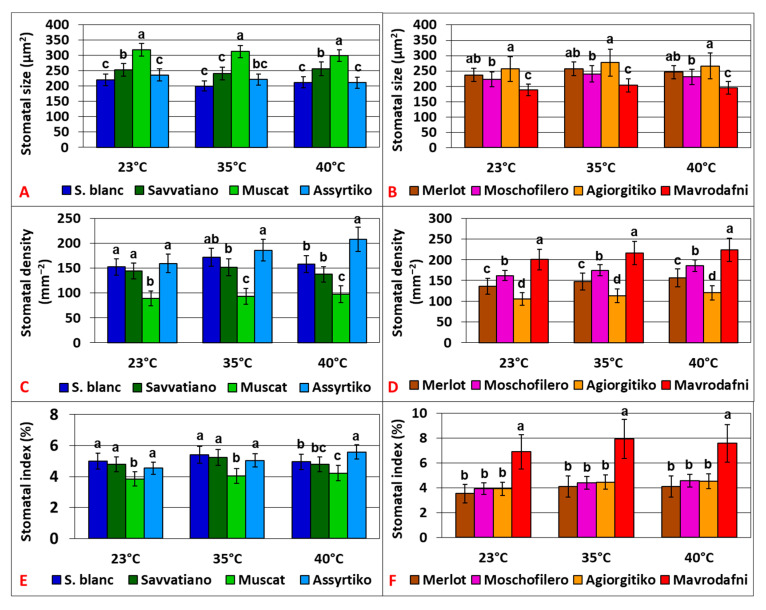
Bar graphs comparing stomatal size (**A**,**B**), density (**C**,**D**), and stomatal index (**E**,**F**) of grapevine cultivars. Error bars represent mean  ±  SD. Different lowercase letters denote statistically significant differences between cultivars (*p* < 0.05). (**A**,**C**,**E**) refer to white cultivars and (**B**,**D**,**F**) to red ones (*n* = 4).

## Data Availability

The original contributions presented in this study are included in the article. Further inquiries can be directed to the corresponding authors.

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
