# Peer review of "Physiological Efficiency and Adaptability of Greek Indigenous Grapevine Cultivars Under Heat Stress and Elevated CO2: Insights into Photosynthetic Dynamics"

_plants, 2025, doi:10.3390/plants14162518_

Round 1
Reviewer 1 Report
Comments and Suggestions for Authors
This study analyzed the physiological efficiency and adaptability of Greek indigenous grapevine cultivars under heat stress and elevated COâ‚‚. The research holds significant implications for understanding the future sustainability of viticulture in the face of climate change. The experiment is well-designed and manuscript is well-written. I did not find any major flaw in the manuscript but some issues require the author's attention before it can be accepted for publication; please revise the article based on the following comments.
- In introduction, authors providing lot of information about increases CO2 concentration but in a repetitive manner that can be consolidated to avoid redundancy. The introduction section does not clearly state the main research gaps in the study. Moreover, authors also do not provide the justification about selection of these specific cultivars. Are these representatives of heat-sensitive varieties? Moreover, sections jump between different concepts without any logical connection. Finally, authors did not provide the general aims of the study without any specific hypothesis.
- The results are overall presented in somewhat clearly and organized manner, with significant differences among cultivars and temperature treatments highlighted effectively.
- The Discussion section is overly simplicity that does not advance our knowledge. Most of the parts are just repeating the results of this study. Authors should provide mechanistic explanations of how elevated CO2 affects photosynthesis and stomatal behavior in more detailed, particularly regarding biochemical processes like RuBP carboxylation and regeneration. The discussion of stomatal strategies (anisohydric vs. isohydric) is insightful, but a more in-depth exploration of how these traits are adaptive in specific grapevine cultivars would enhance the relevance. Additionally, a clearer link between WUE, stomatal conductance, and temperature dependence would strengthen the interpretation. Finally, suggesting future research directions would strengthen the discussion.
- Line 371-373: The authors continuously used the term "international cultivars"? What is the specific aim behind using this term? Using the term "cultivars" is not enough?
- Line 466-468: There is contradiction between heading and description. In heading you are specifying about anatomical traits and then you describe in the text that leaf anatomical traits. Clarify it.
- Line 482-489: The authors performed combination of ANOVA, Tukey’s HSD, T-tests, and regression analyses. The authors should clarify the purpose of each test to avoid the redundant tests. For example: If T-tests are used, it should be stated clearly why they are needed after an ANOVA, and whether they are testing comparisons not already handled by the Tukey’s post-hoc test. The authors also did not specify either performed independent sample T-test or paired T-test. Also clarify about dependent and independent variables in statistical analysis.
- Line 490: The conclusion is about to conclude the main and important findings of study and what are their implications. This section is somewhat scattered and difficult to follow up. Authors start the last sentence with “Findings”. Which findings? It’s vague.
Author Response
Reviewer 1
“This study analyzed the physiological efficiency and adaptability of Greek indigenous grapevine cultivars under heat stress and elevated COâ‚‚. The research holds significant implications for understanding the future sustainability of viticulture in the face of climate change. The experiment is well-designed and manuscript is well-written. I did not find any major flaw in the manuscript but some issues require the author's attention before it can be accepted for publication; please revise the article based on the following comments.”
Response:
Thank you very much for your positive comments and also for the targeted and constructive suggestions. We took into consideration all your suggestions and the manuscript has been revised accordingly.
Comment 1:
“In introduction, authors providing lot of information about increases CO2 concentration but in a repetitive manner that can be consolidated to avoid redundancy. The introduction section does not clearly state the main research gaps in the study. Moreover, authors also do not provide the justification about selection of these specific cultivars. Are these representatives of heat-sensitive varieties? Moreover, sections jump between different concepts without any logical connection. Finally, authors did not provide the general aims of the study without any specific hypothesis.”
Response:
Based on your comments, we revised and shortened the information about increases of CO2 concentration in the Introduction section to avoid redundancy (lines 38-40).
To highlight the gaps in the literature the present study aims to cover, we better emphasize that the relative studies are quite limited, particularly those dealing with the responses of grapevine cultivars of Greek origin (lines 62-69).
The varieties selected for this study were chosen based on their high viticultural and oenological potential, as well as their origin from the southern mainland and insular regions of Greece, a plausible indication of increased adaptation to dry-thermal conditions, with potential contribution to addressing climate change resilience. This important information has been now included in lines 99-102.
We agree with your comment that in parts there was a lack of logical connection between different concepts. Therefore, we implemented the necessary revisions to improve coherence in the Introduction section (for instance, lines 56-58 and 70-72).
Finally, a small paragraph has been adder at the end of the Introduction section and to better show the aims and overarching goals of the study (Lines 97-105).
Comment 2:
The results are overall presented in somewhat clearly and organized manner, with significant differences among cultivars and temperature treatments highlighted effectively.
Response:
We sincerely thank the reviewer for their positive and encouraging comments.
Comment 3:
The Discussion section is overly simplicity that does not advance our knowledge. Most of the parts are just repeating the results of this study. Authors should provide mechanistic explanations of how elevated CO2 affects photosynthesis and stomatal behavior in more detailed, particularly regarding biochemical processes like RuBP carboxylation and regeneration. The discussion of stomatal strategies (anisohydric vs. isohydric) is insightful, but a more in-depth exploration of how these traits are adaptive in specific grapevine cultivars would enhance the relevance. Additionally, a clearer link between WUE, stomatal conductance, and temperature dependence would strengthen the interpretation. Finally, suggesting future research directions would strengthen the discussion.
Response:
Thank you for your useful suggestions. Following your instructions, we added appropriate mechanistic explanations of how elevated CO2 affects stomatal behavior and photosynthesis in lines 240-245, 262-265, 269-271, and 314-318.
We have expanded our discussion on how different stomatal strategies relate to the long-term adaptation of grapevine varieties to varying climatic conditions. We also included examples of specific cultivars, both from our study and from other existing research (lines 351-363). Two new citations have been added [70 and 71].
Furthermore, we included an explanation about the relationship between WUE, stomatal conductance, and their temperature dependencies in lines 298-302.
Finally, we added suggestions for future research in lines 535-539.
Comment 4:
Line 371-373: The authors continuously used the term "international cultivars"? What is the specific aim behind using this term? Using the term "cultivars" is not enough?
Response:
We agree and the term “international varieties” was removed from both the Abstract and the Materials & Methods sections.
Comment 5:
Line 466-468: There is contradiction between heading and description. In heading you are specifying about anatomical traits and then you describe in the text that leaf anatomical traits. Clarify it.
Response:
Thank you for your comment. We have now resolved this contradiction by modifying the title and text descriptions as well (lines 486-487, and 495)
Comment 6:
Line 482-489: The authors performed combination of ANOVA, Tukey’s HSD, T-tests, and regression analyses. The authors should clarify the purpose of each test to avoid the redundant tests. For example: If T-tests are used, it should be stated clearly why they are needed after an ANOVA, and whether they are testing comparisons not already handled by the Tukey’s post-hoc test. The authors also did not specify either performed independent sample T-test or paired T-test. Also clarify about dependent and independent variables in statistical analysis.
Response:
Following your suggestion, we clarified the distinct purpose of each test to avoid redundancy. Specifically, for the paired samples T-tests, we clarified that they were used separately to assess the effect of CO2 concentration on the dependent variables (An, gs, and WUE), as the independent variable (CO2) had only two levels (400 ppm and 700 ppm) making the use of ANOVA and Tukey’s HSD post-hoc test inapplicable (lines 506-509). We also specified that we performed paired samples T-tests. Finally, we have specified in parentheses the independent variables (variety and temperature) and the dependent variables (An, gs, WUE, Vcmax, Jmax, Amax, Ls, Lm, Lb, stomatal size/density and SI) in lines 503-505.
Comment 7:
Line 490: The conclusion is about to conclude the main and important findings of study and what are their implications. This section is somewhat scattered and difficult to follow up. Authors start the last sentence with “Findings”. Which findings? It’s vague.
Response:
We agree. Following your valuable suggestions, we have incorporated the main findings of this study along with their implications. Additionally, we revised the whole “Conclusions” section to improve readability and clarity.
Reviewer 2 Report
Comments and Suggestions for Authors
Dear Authors
This study investigates the impact of climate change on key physiological parameters of Greek indigenous grapevine cultivars (Savvatiano, Muscat, Assyrtiko, Mavrodafni, Moschofilero, and Agiorgitiko), using international varieties (Sauvignon blanc and Merlot) as benchmarks. The aim was to identify genotypes with higher photosynthetic dynamics and water use efficiency (WUE) under heat stress and to examine the role of COâ‚‚ enrichment in modulating these responses.
Introduction:
Briefly specify what is already known about the physiological responses of these Greek cultivars and what is missing.
lines 47-49 "Given this scenario, major devastating consequences are expected to affect viticulture worldwide, which could be addressed by redefining suitable wine-growing regions, adopting sustainable viticultural practices, or cultivating heat-resistant cultivars [5-6]"
[5] Lippi, P., Mattii, G. B., & Cataldo, E. (2025). Biochar, Properties and Skills with a Focus on Implications for Vineyard Land and Grapevine Performance. Phyton-International Journal of Experimental Botany, 0-0. [6]Cataldo, E., Fucile, M., & Mattii, G. B. (2022). Composting from organic municipal solid waste: a sustainable tool for the environment and to improve grape quality. The Journal of Agricultural Science, 160(6), 502-515.lines 78-79 I suggest the authors to discriminate between isohydric and anisohydric cultivars
line 87 Specify which WUE you are talking about
M&M
Specify the number of biological replicates per cultivar/treatment. Specify the phenological periods of the surveys.
Indicate whether the assumption of normality and homoscedasticity was tested before the ANOVA.
Discussion
Very good, but I suggest:
More conciseness in the subsections: many discussions are long and redundant. Sections 3.1 and 3.2 in particular could be more concise.
Curiosity: It would be interesting to know if the differential responses between varieties are consistent with their range of origin (climatically).
Bibliography
Suggestion: Standardize article formatting (some have DOIs, others do not).
Author Response
Reviewer 2
Dear Authors
This study investigates the impact of climate change on key physiological parameters of Greek indigenous grapevine cultivars (Savvatiano, Muscat, Assyrtiko, Mavrodafni, Moschofilero, and Agiorgitiko), using international varieties (Sauvignon blanc and Merlot) as benchmarks. The aim was to identify genotypes with higher photosynthetic dynamics and water use efficiency (WUE) under heat stress and to examine the role of COâ‚‚ enrichment in modulating these responses.
Introduction:
Comment:
Briefly specify what is already known about the physiological responses of these Greek cultivars and what is missing.
Response:
We sincerely thank you for your helpful advice. Following your instructions, we added this information in lines 62-69.
Comment:
lines 47-49 "Given this scenario, major devastating consequences are expected to affect viticulture worldwide, which could be addressed by redefining suitable wine-growing regions, adopting sustainable viticultural practices, or cultivating heat-resistant cultivars [5-6]"
[5] Lippi, P., Mattii, G. B., & Cataldo, E. (2025). Biochar, Properties and Skills with a Focus on Implications for Vineyard Land and Grapevine Performance. Phyton-International Journal of Experimental Botany, 0-0.
[6]Cataldo, E., Fucile, M., & Mattii, G. B. (2022). Composting from organic municipal solid waste: a sustainable tool for the environment and to improve grape quality. The Journal of Agricultural Science, 160(6), 502-515.
Response:
Thank you for suggesting these relevant citations. We have incorporated these two references into the revised manuscript (lines 41-44; # 3, 4 in References).
Comment:
lines 88-89 I suggest the authors to discriminate between isohydric and anisohydric cultivars
Response:
We have now discriminated isohydric and anisohydric cultivars based on their scientific definitions (lines 93-96).
Comment:
line 87 Specify which WUE you are talking about
Response:
We have specified that we refer to intrinsic water use efficiency (iWUE) that is defined as An/gs (lines 89-90). It has also been changed throughout the manuscript.
M&Ms
Comment:
Specify the number of biological replicates per cultivar/treatment. Specify the phenological periods of the surveys.
Response:
Many thanks for your comment. We have now specified the number of biological replicates per cultivar/treatment (n=4) both in Materials and Methods (lines 398-401) and in the Figure legends. The surveys were conducted in a growth chamber under controlled conditions and during measurements the phenological period was approximately from flowering to the onset of ripening (stages 19-45 in the Eichhorn-Lorenz scale). However, no clusters were left in the vines during the experiments.
Comment:
Indicate whether the assumption of normality and homoscedasticity was tested before the ANOVA.
Response:
Thank you for your helpful guidance. Indeed, the assumption of normality and homoscedasticity was tested before the ANOVA. Following your suggestion, we added this information in lines 506-508.
Discussion
Comment:
Very good, but I suggest:
More conciseness in the subsections: many discussions are long and redundant. Sections 3.1 and 3.2 in particular could be more concise.
Response:
We appreciate your kindness and your encouraging comments.
The whole Discussion section has been revised and is now more concise. The sub-sections 3.1 and 3.2, in particular, have been extensively revised to avoid redundancy.
Comment:
Curiosity: It would be interesting to know if the differential responses between varieties are consistent with their range of origin (climatically).
Response:
Good point. Although the number of cultivars is limited and the range of climate conditions in the respective regions is rather narrow, it is noteworthy that there was not a clear association between a cultivar’s performance under heat stress and the climate of its region of origin. For instance, although Assyrtiko that exhibited the best performance originates from an island with a hot and dry climate, Savvatiano, a variety native to Attica (around Athens), performed similarly to Sauvignon blanc. This has been now stated in lines 361-363-361 & 527-530.
Comment:
Bibliography
Suggestion: Standardize article formatting (some have DOIs, others do not
Response:
We have now corrected any imperfections and standardized the formatting of the references, based also on the Journal’s requirements.
Reviewer 3 Report
Comments and Suggestions for Authors
This is useful and interesting examination of combined elevated CO2 and temperature on grape leaf physiology, comparing locally adapted and other cultivars.
My main concern is a need for some clarification about the experimental design: how many plant growth chambers were used? There were 6 different growth conditions (3 temperature regimes, each with 2 CO2 treatments), and eight cultivars were exposed to each, yet it sounds as if only one chamber was used. That could be done with one chamber, but more likely was done using several. Were all cultivars grown simultaneously in the same chamber, or sequentially? I did not see how many individual plants were measured for each cultivar - growth condition combination. Standard deviations were given in the figures, but I did not see an "n =" statement with these.
It was striking to me how low stomatal conductances were at 23 C compared to the higher temperatures. Is that normal for grapes? The small effect of growth CO2 on gs was striking - are there other similar data for grapes?
Author Response
Reviewer 3
This is useful and interesting examination of combined elevated CO2 and temperature on grape leaf physiology, comparing locally adapted and other cultivars.
Response:
Thank you very much for your constructive and encouraging comments. Below we provide responses to the points you raised, with the aim of further clarifying the ambiguities and improving the formulation of the paper.
Comment
My main concern is a need for some clarification about the experimental design: how many plant growth chambers were used? There were 6 different growth conditions (3 temperature regimes, each with 2 CO2 treatments), and eight cultivars were exposed to each, yet it sounds as if only one chamber was used. That could be done with one chamber, but more likely was done using several. Were all cultivars grown simultaneously in the same chamber, or sequentially? I did not see how many individual plants were measured for each cultivar - growth condition combination. Standard deviations were given in the figures, but I did not see an "n =" statement with these.
Response:
Just one growth chamber was used in the experiment, in which each group of cultivars (white or red) was cultivated separately. Specifically, each group of cultivars was firstly exposed to control temperature (CT), then to moderate heat stress (MHS), and finally to severe heat stress (SHS), with 400 ppm applied first and 700 ppm thereafter at each temperature regime. Each group of varieties consisted of a total of 16 plants, corresponding to 4 plants per variety (n=4). We have now added the number of replicates to Figure legends, and all this information to Materials and Methods (lines 398-404).
Comment
It was striking to me how low stomatal conductances were at 23ËšC compared to the higher temperatures. Is that normal for grapes? The small effect of growth CO2 on gs was striking - are there other similar data for grapes?
Response:
We thank you for this observation. The differences in gs between 23°C and 35°C were not equally pronounced across all cultivars, with the most notable increases observed in Mavrodafni and Sauvignon blanc. Although there is not much information available in the literature, similarly large differences of temperature-induced gs have been recorded in Sangiovese [63] and Merlot [64], where gs remained below 100 mmol m-2 s-1 at temperatures close to 20°C and exceeded 300 mmol m-2 s-1 at 30-35°C. This has been now discussed in lines 338-340.
Here we also observed that elevated CO2 (e[CO2]) led to reduction of gs up to 26.4% in red cultivars and up to 27.3% in white cultivars. Similarly, Schultz and Stoll [45], referred that gs in grapevine cultivars declined upon rising COâ‚‚, exhibiting a typical average decrease of about 20%. Leibar et al. [46], reported that well irrigated vines (cv. Tempranillo) subjected to two different CO2 concentrations (375 ppm vs 700 ppm) exhibited CO2-mediated gs decrease of 19%. In other studies (e.g., Moutinho-Pereira et al. 2009, Vitis 48 (4), 159-165) gs remained almost unchanged in response to e[CO2]. This information is now presented in lines 271-273.
Round 2
Reviewer 2 Report
Comments and Suggestions for Authors
Accept in present form